# Predictors of Influenza Vaccination Uptake and the Role of Health Literacy among Health and Social Care Volunteers in the Province of Prato (Italy)

**DOI:** 10.3390/ijerph19116688

**Published:** 2022-05-30

**Authors:** Chiara Lorini, Vieri Lastrucci, Beatrice Zanella, Eleonora Gori, Fabrizio Chiesi, Angela Bechini, Sara Boccalini, Marco Del Riccio, Andrea Moscadelli, Francesco Puggelli, Renzo Berti, Paolo Bonanni, Guglielmo Bonaccorsi

**Affiliations:** 1Department of Health Sciences, University of Florence, Viale GB Morgagni 48, 50134 Florence, Italy; chiara.lorini@unifi.it (C.L.); vieri.lastrucci@meyer.it (V.L.); angela.bechini@unifi.it (A.B.); sara.boccalini@unifi.it (S.B.); paolo.bonanni@unifi.it (P.B.); guglielmo.bonaccorsi@unifi.it (G.B.); 2Epidemiology Unit, Meyer Children’s Hospital, 50139 Florence, Italy; 3Medical Specialization School of Hygiene and Preventive Medicine, University of Florence, 50134 Florence, Italy; eleonora.gori@unifi.it (E.G.); marco.delriccio@unifi.it (M.D.R.); andrea.moscadelli@unifi.it (A.M.); 4Central Tuscany Local Health Unit, 50142 Florence, Italy; fabrizio.chiesi@gmail.com; 5Management Department, Meyer Children’s University Hospital, 50139 Florence, Italy; francesco.puggelli@meyer.it; 6Azienda Sanitaria Locale Toscana Centro, Department of Prevention, Via Lavarone 3/5, 59100 Prato, Italy; renzo.berti@uslcentro.toscana.it

**Keywords:** influenza, vaccine, volunteers, survey, Italy, COVID-19

## Abstract

Annual influenza vaccination is recommended for volunteers involved in primary health and social services. Little is known about the volunteers’ adhesion to influenza vaccination recommendations. The aim of this study was to assess influenza vaccination determinants among a group of volunteers who provided essential activities during the first SARS-CoV-2 pandemic wave in the province of Prato, Tuscany (Italy) and to evaluate the role of health literacy in influencing vaccination determinants. Method: In this cross-sectional study, the predictors of influenza vaccination uptake were assessed through the administration of a questionnaire. Variables significantly associated with influenza vaccination uptake were included in five multivariate logistic regression models through a backward stepwise procedure. Results: Among the 502 enrolled volunteers, 24.3% reported being vaccinated in the 2019–2020 season. Vaccination uptake was 48.8% in participants aged 65 years or older and 15.7% in those aged 64 years or younger. Considering the whole sample in the final model of multivariate logistic regression analysis, the predictors of influenza vaccination uptake were age (OR = 1.05; 95% CI = 1.03–1.07), presence of heart diseases (OR = 2.98; 95% CI = 1.24–7.19), pulmonary diseases (OR = 6.18; 95% CI = 2.01–19.04) and having undergone surgery under general anesthesia in the prior year (OR = 3.14; 95% CI = 1.23–8.06). In the multivariate model considering only participants with a sufficient level of health literacy (HL), none of these predictors resulted in significant associations with vaccination uptake, except for age (OR= 1.04; 95% CI = 1.02–1.07). Conclusions: Our findings revealed a very low influenza vaccination uptake among volunteers, suggesting the need to increase awareness in this at-risk group by means of a better communication approach.

## 1. Introduction

Influenza is a contagious respiratory illness caused by an RNA virus of the Orthomyxoviridae family and causing mild-to-severe illness. Serious outcomes can result in hospitalization or death. Some people, such as older people, young children, and people with chronic medical conditions, are at high risk of serious complications [1]. The disease burden for influenza is high: globally the annual epidemics results in about 3–5 million cases of severe illness and about 290,000–650,000 deaths [2,3]. In Europe, in the period 2009–2013, influenza had the highest burden among 31 different infectious diseases, with a disability-adjusted life years per 100,000 population of 81.8 [4]. Nevertheless, vaccination coverage rates (VCRs) are still low/suboptimal in most countries and far below the recommended rate of 75% in most instances [5,6,7,8,9,10,11,12,13].

Influenza vaccination uptake may depend on several factors, such as social determinants (e.g., age, gender, socio-economic status, etc.), intermediary determinants (e.g., residential location, behavioral beliefs, sources of information, etc.), and welfare system-related factors, as well as on perceptions of vaccine efficacy, safety, and adverse events [14,15,16,17,18,19]. Health Literacy (HL), defined as “*people’s knowledge, motivation and competences to access, understand, appraise, and apply health information in order to make judgments and take decisions in everyday life concerning healthcare, disease prevention and health promotion to maintain or improve quality of life during the life course*” [20], has been reported to influence several health outcomes and behaviors [21,22,23,24], and recently, it was hypothesized that HL may be a determinant of vaccination decision-making [25,26].

Participants involved in primary social and health-care services can greatly benefit from vaccination. Indeed, in Italy, influenza vaccination is recommended to this group [27]. In Italy, volunteering exerts a fundamental role in supporting these services, and vaccination is strongly recommended and offered free of charge to volunteers in the primary health and social services (VPHSS). Voluntary organizations are bodies involved in different activities of general interest that assist people in need through the voluntary service of their members. These organizations were introduced in the Italian system by Law 266/1991 and were subsequently recognized as non-profit organizations [28]. About 10.7% of the Italian general population aged more than 14 years is involved in these activities, some of which are within the national health-care system perimeter, such as health-care support (mainly at patients’ homes) and health emergency transport. These activities may expose volunteers to a higher risk of contracting and transmitting the influenza virus. During the SARS-CoV-2 pandemic waves, in particular during the first phase, volunteers were widely involved in the management of the COVID emergency, such as in the hospital transfer of patients with severe symptoms that required hospitalization, or in the delivery of first-aid care and goods (drugs or food) to those who were quarantined.

To the best of our knowledge, despite the many functions performed for the benefit of public health-care and social services, scarce attention in the literature has been devoted to influenza vaccination uptake and its determinants in volunteers. Thus, there is the need to acquire more information about this topic. The aim of this study was, therefore, to assess influenza vaccination predictors (sociodemographic data, health literacy, comorbidity, type of employment) among a group of volunteers who provided essential activities supporting health-care services during the first pandemic period in the province of Prato (Tuscany, central Italy). Furthermore, the study evaluated whether health literacy levels play a role in influencing the predictors of influenza vaccination uptake.

## 2. Materials and Methods

This cross-sectional study was carried out to assess the predictors of influenza vaccination uptake in a population of volunteers through the administration of a questionnaire from April to June 2020. The population was composed of volunteers enrolled in a sero-epidemiological study for the assessment of the spread of COVID-19 [29].

The study was approved by the Ethics Committee of the Area Vasta Toscana Centro (Comitato Etico Regionale per la Sperimentazione Clinica della Regione Toscana, Sezione Area Vasta Centro, Florence, Italy, 17470_oss) and was conducted according to the Declaration of Helsinki.

### 2.1. Study Setting, Population, and Questionnaire

In Italy, for the 2019–2020 season and the seasons that followed, influenza vaccination was recommended and free of charge for: (i) older adults aged ≥65 years; (ii) people aged 6 months to 64 years affected by at-risk health conditions (e.g., chronic respiratory and cardiovascular pathologies, cancer, other forms of immunodeficiency); (iii) specific categories of professionals who are engaged in activities in the public interest (e.g., health-care workers, volunteers in the health and social services); and (iv) other categories (e.g., veterinarians, pregnant women, blood donors) [30,31,32].

The study was carried out in a population-based sample composed of participants who belonged to one of the different volunteer associations (such as “Civil Protection” or “Misericordia”) which provided essential health and social support activities in the Province of Prato (Tuscany, Italy) during the first pandemic wave and, specifically, in the general lock-down phase in 2020 (March to May 2020). 

All the volunteer associations working in the Province of Prato participated in the study, and all the volunteers were invited to participate in the study after signing an informed consent form; the only inclusion criteria were that participants were required to be aged ≥18 years old and to join and give written consent. No sampling procedure was applied since all the volunteers who fulfilled the inclusion and exclusion criteria were invited to participate.

The enrolled participants were asked to fill out a questionnaire divided into different sections related to (Appendix A):sociodemographic data: sex, age, nationality, educational level, type of employment (the “not employed” group included unemployed people, housewives, students, and retired people);health literacy: measured with the HLS-EU-Q6 (further description is provided below);living conditions: living with people aged >64 years old or with people with chronic diseases or suffering from immunodeficiency;at least one of the following risk conditions or diseases: diabetes, obesity, heart diseases, pulmonary diseases, diseases of the immune system, chronic kidney diseases, chronic liver diseases, organ or bone marrow transplants, chronic neurological diseases, oncological diseases (last 5 years), hematological diseases, pregnancy, or surgery under general anesthesia (in the previous year);smoking habits: never smokers, former smokers, current smokers (fewer than 10 cigarettes/day, 10–20 cigarettes/day, more than 20 cigarettes/day);influenza vaccination: one multiple-choice question about having received influenza vaccination during the 2019–2020 influenza epidemic season (yes, no, I do not remember).

The HL level was assessed using the Italian version of the 6-item European Health Literacy Survey Questionnaire (HLS-EU-Q6), which is the short-short form of the 47-item tool (HLS-EU-Q47) [33,34]. It is a self-report instrument with Likert-type responses (“very easy”, “fairly easy”, “fairly difficult”, “very difficult”) and generates a final score that can be used to measure HL in general populations. For each item, the following scores were considered: “very easy” = 4; “fairly easy” = 3; “fairly difficult” = 2; “very difficult” = 1. “Don’t know” or refusal were recorded as missing responses. The final scale score for the survey was the mean value and varied between 1 and 4. Only respondents who answered at least five items were considered. According to the final score, three possible levels of HL were defined: inadequate HL (score ≤2); problematic HL (score ranged from 2 to 3); and sufficient HL (score ≥3). The HLS-EU-Q6 is considered an economic measure of HL to be included in surveys where the measurement of HL is not the main aim [35].The Tuscany Region has used the HLS-EU-Q6 for Italian lifestyle surveillance systems PASSI (progress by local health units towards a healthier Italy) since 2017 [33,34] and the instrument has proved to be an effective measure of HL in the context of the general population.

### 2.2. Statistical Analysis

Answers were collected and entered into a database and subsequently analyzed using IBM SPSS 27.0 (IBM, Armonk, NY, USA).

The enrolled participants were assigned to different groups according to (i) sociodemographic information, (ii) HL level, (iii) risk conditions or diseases, and (iv) smoking habits. A descriptive analysis was performed to evaluate the frequencies and the percentages of the collected answers and to assess vaccination uptake in the 2019–2020 season related to the different groups of the study population (categorical and numerical variables). Fisher’s exact test and Mann–Whitney test for independent samples were used to assess significant differences in the answers according to the different categorical variables and numerical variables, respectively. 

Variables significantly associated with influenza vaccination uptake were included in five multivariate logistic regression models through the backward stepwise procedure in order to calculate the odds ratio for being vaccinated against influenza. In particular, the five models were fitted in order to identify vaccination uptake predictors in the following population groups: the whole sample; people ≤64 years old; people >64 years old; people with sufficient health literacy (HL) according to HLS-EU-Q6; and people with problematic or inadequate health literacy (HL) according to HLS-EU-Q6. A *p*-value less than or equal to 0.05 was considered statistically significant.

## 3. Results

A total of 502 volunteers agreed to participate and filled in the questionnaire. The participation rate was 95.5%. 

The descriptive analysis of the collected data is reported in Table 1 and Table 2, for the whole sample and by influenza vaccination uptake, respectively. Most of the volunteers were males (65.1%), younger than 65 years old (75.9%; median age of 53 years old), Italian (97.8%), and not employed (60.2%). Slightly less than half of the sample (48%) had a high school diploma or a university degree (38.6% and 9.4%, respectively).

HL was measured for 86.3% of the sample and the median score for the HLS-EU-Q6 was 3; among the respondents, 50.8% presented sufficient HL, while 35.5% were categorized as having inadequate or problematic HL.

Considering living conditions, 29.7% lived with people older than 64 years old or with chronic diseases. The median number of cohabitants was 3, of rooms in the house, 4, and of bedrooms, 2. As for risk conditions or diseases with complications of concern for influenza, the more frequently reported were obesity (7%), heart diseases (5%), diabetes (4.6%) and pulmonary diseases (3.4%); as a whole, 26.7% of the sample reported having at least one of the considered risk conditions or diseases.

About half of the sample (50.6%) were never smokers, while 20.3% were former smokers.

Volunteers who reported being vaccinated in the 2019–2020 season accounted for 24.3% of the participants. Regarding socio-demographic data, influenza vaccination uptake was significantly associated with age (higher in older people), educational level (higher in the less educated), employment (higher in the not employed), and number of cohabitants (vaccinated people presented a lower median value for cohabitants). Specifically, considering age, the percentage of vaccinated people was higher among people older than 64 years. Excluding older people (>64 years old), vaccinated volunteers were still significantly older than the non-vaccinated (median age: 52.5 and 44 years, respectively). Moreover, vaccination uptake was significantly higher in people with diabetes (52.2%), heart diseases (60%), pulmonary diseases (52.9%), among those who had oncological diseases in the previous 5 years (60%), surgery under general anesthesia in the previous year (43.5%), and those who indicated having at least one of the considered risk conditions or diseases (38.8%). On the contrary, influenza vaccination uptake was not significantly associated with sex, HL (either considering the level or the score on the HLS-EU-Q6), smoking habits, number of rooms or bedrooms in the house, and the other single-risk conditions of diseases (Table 2).

Variables significantly associated with influenza vaccination uptake were included in five multivariate logistic regression models. In Table 3, the final models are reported, namely those in which the variables with no significant associations were excluded using the backward stepwise procedure. Considering the whole sample, the predictors of influenza vaccination uptake were age (OR = 1.05), suffering from heart diseases (OR = 2.98) or pulmonary diseases (OR = 6.18), and having undergone surgery under general anesthesia in the previous year (OR = 3.14). In the younger subgroup (≤64 years old), predictors were the same as those in the whole group, although they exhibited higher ORs and included having diabetes (OR = 4.9) and oncological diseases in the prior five years (OR = 5.30). In contrast, for the older subgroup (>64 years old), having at least one of the listed risk condition or diseases was the only predictor (OR = 3.22). The predictors differed also considering two subgroups according to HL level. While age was the only variable that remained in the final model among volunteers with sufficient HL (OR = 1.04), among those with inadequate or problematic HL, the predictors were age (OR = 1.05), diabetes (OR = 6.34), heart diseases (OR = 4.53), pulmonary diseases (OR = 8.73), and having undergone surgery under general anesthesia in the prior year (OR = 6.38).

## 4. Discussion

This study aimed to identify the individual predictors of influenza vaccination uptake in a sample of volunteers who were involved in essential activities supporting health and social services during the first period of the COVID-19 pandemic (March–April 2020). Furthermore, the study aimed to explore the role of health literacy in influencing the identified predictors of influenza vaccination uptake. The considered predictors included sociodemographic characteristics, living conditions, risk conditions or diseases, and smoking habits. 

### 4.1. Theoretical and Practical Implications

The overall influenza vaccination uptake was about 25% in the whole sample and about 50% in participants aged 65 years and older. Regarding the presence of concomitant health conditions, the highest percentages of vaccinated participants were among those who suffered from diabetes, heart diseases, or an oncological disease in the prior five years, pulmonary diseases, and those having undergone surgery under general anesthesia in the prior year; these results are in line with the national recommendations for flu vaccination in high-risk groups [30,31,32].

Our study population represents the national structure of the voluntary associations well, which are mainly composed of males, those younger than 54 years old, those with a higher educational degree (high school diploma or university degree), and those who are not employed (students, housewives, the retired, or those looking for a job) [36]. The fields of intervention for the voluntary associations include health care, social welfare, and civil protection [37,38]. For the 2019–2020 influenza season in Italy, influenza vaccination was recommended for the elderly (people aged ≥65 years), pregnant women, people living with chronic conditions, people at high risk of professional exposure (such as health-care professionals), and people involved in public services of primary collective interest. Among these last, the volunteers are also included, in particular those who offer health-care support [32]. Volunteers involved in health-care services may have a role comparable to the one exerted by the health-care and social care personnel, and thus, may be exposed to the same risks. As a matter of fact, the literature highlighted an increased risk for influenza infections and diseases among health workers [39]. Italy is not the only country that provides this recommendation. Other nations, for instance, Germany, the United States and Canada, also recommend influenza vaccination for volunteers or those in services dealing extensively with the public [40,41,42]. The volunteers’ rate for vaccination uptake that we found (24.3% for the whole sample and 48.8% in the aged >64 group) seems to be quite in line with Italian influenza VCR for 2019–2020 flu season: 16.8% for the general population and 54.6% for older people [43]. Taking into account previous influenza seasons (from 2010–2011 and 2018–2019), the VCR in Tuscany ranged from 16.5% to 22.5% for the general population and from 49.9% to 68.8% among people aged >64 years [44]. These data highlight that adherence to vaccination recommendations is far below the minimum targets, set at 75% for high-risk groups. Our overall influenza vaccination adherence rate (24.3%) is lower than those found in similar cross-sectional studies involving general adult populations carried out in the US (42.3%) [45], in the city of Tokyo (38.1%) [46], and comparable to that assessed in a cross-sectional study carried out in Singapore in 2013 involving adults aged ≥50 years (15.2%) [47].

The multivariate analysis performed in our study found age, the presence of heart and pulmonary diseases, and having undergone surgery as predictors of influenza vaccination uptake for the whole sample. Many studies reported age [45,48,49,50,51,52,53] and presence of a chronic condition at risk for influenza [46,47,49,50,51] as predictors of flu vaccination uptake. On the other hand, in our study, sex, number of cohabitants, and employment status were not associated with vaccination uptake. In the literature, conflicting results on the roles of these factors have been reported [47,52,54,55].

As far as age and vaccination uptake are concerned, our VCR in participants aged 65 years or older is in line with other studies [48,49]. A positive correlation between age and vaccination uptake was expected since there is a national recommendation based on age, and moreover, vaccination is actively offered by general practitioners to all people aged 65 years or older. However, the large difference (an almost 3-fold difference) in VCR between volunteers aged 65 years and older (48.8%) and those younger than 65 years (15.7%) suggests that volunteers tend to be vaccinated more for their individual demographic conditions (i.e., older age) than for their occupational exposure (i.e., being a volunteer involved in primary health services). Therefore, it seems that age as a risk factor for influenza is a well-known concept among volunteers, while, on the contrary, the risks derived from volunteering activities are recognized and considered less.

Considering the published research, it is still not clear what the effect of HL is on influenza vaccination uptake [56]. Some studies suggest that low levels of HL are associated with lower influenza uptake [25,57,58,59]. On the other hand, no associations were found in specific groups of the population, such as non-familial, paid caregivers or nursing home staff [60,61]. In our study population, we did not find a significant association between HL and influenza vaccination uptake. However, it seems that HL level could influence the role of the identified predictors of vaccination uptake. While in the problematic or inadequate HL groups, several predictors of vaccination uptake did emerge, none of the identified factors predicted vaccination uptake in the participants with sufficient HL (with the exception of age). These findings suggest that a higher level of HL may reduce the role of other predictors of vaccination uptake, and this effect of HL may be explained by the fact that those with high HL levels are more aware of the benefits of vaccination; this high level of awareness may also mitigate any additional effects provided by other factors, such as having at-risk conditions for influenza. Specific competences in vaccination, especially vaccine literacy, need to be examined as potential predictors of influenza vaccination uptake among volunteers in future research.

### 4.2. Strengths and Limitations of the Study; Future Perspectives

The present study has several strengths and some limitations. As for the strengths, this is one of the first studies assessing predictors of influenza vaccination uptake in volunteers. Furthermore, the study explored the role of health literacy in vaccination uptake, a topic that only recently has gained attention in the literature. Lastly, the study population can be considered representative of the entire study area for the selected population groups. Moreover, the enrolled participants represent the structure of the voluntary associations in Italy well, so we may be able to infer some information of general interest regarding volunteers.

As for the study limitations, first, the data were self-reported by the participants, and therefore, the results may have suffered from the social desirability bias of the participants, especially in reporting their vaccination status. However, it should be emphasized that the survey was self-administered and completely anonymous, and this may have limited the possible social desirability biases. Moreover, since no objective evaluations or checks were performed, the collected data could have been affected by recall bias. This aspect may have mainly concerned vaccination uptake by people who are vaccinated occasionally (i.e., not every year). Future studies will be performed in order to compare self-reported versus objective influenza vaccination uptake among volunteers. 

Furthermore, HL was measured using a self-reported instrument of perceived difficulties in performing different health tasks, so overconfidence or lack of confidence could have led to the underestimation or overestimation of health literacy. People tend to be overconfident or lack confidence as a consequence of the connection between knowledge, confidence, self-efficacy, and emotional distress [62]. Since overconfidence and lack of confidence are influenced by cultural and demographic factors [21], we can assume that they may generate biases in estimating HL level and presumably in assessing its relationship with other data. Nonetheless, in our opinion, the use of a self-assessed rather than performance-based measure of health literacy allows us to evaluate the balance between individual skills and the demands and complexities of societal systems, which is the real essence of HL research.

Finally, the study did not consider the exact job duties performed by volunteers; thus, further research considering this aspect is warranted as job duties may play a role in influencing risk perception and vaccination uptake.

## 5. Conclusions

Volunteers involved in health or social services are at an increased risk of contracting influenza, and, at the same time, represent a risk of spreading the virus to the fragile people to whom they offer their services. This cross-sectional study described, for the first time, influenza vaccination uptake and its related predictors in a group of volunteers involved in essential activities during the first wave of the COVID-19 pandemic in the Province of Prato. We found a low overall influenza vaccination uptake; moreover, age and several risk conditions were associated with higher vaccination uptake among volunteers. Lastly, a high level of health literacy seems to mitigate the effects of the identified predictors, probably due to an augmented level of awareness of the benefits of vaccination. Our results could be useful to health authorities and policy makers in order to strengthen the recommendations for influenza vaccination in this population group. From this perspective, the routine surveillance of vaccination coverage among volunteers should be encouraged. Moreover, the findings also suggest the importance of increasing awareness in this specific population group through a better approach to communication in order to increase their adherence to vaccination recommendations and protect themselves as well as the frail people who are the beneficiaries of their services.

## Figures and Tables

**Table 1 ijerph-19-06688-t001:** Descriptive analysis of the categorical variables in the whole sample and by influenza vaccination uptake in the 2019–2020 season. NOTE: °: row percentage; *: Fisher’s exact test.

Variables		N (%)	Influenza Vaccination		
			Yes N (%) °	No/Don’t Remember N (%) °	*p* *
			122 (24.3%)	380 (75.7%)	
Sex	Females	175 (34.9%)	35 (20%)	140 (80%)	0.100
	Males	327 (65.1%)	87 (26.6%)	240 (73.4%)	
Age	≤64 years	381 (75.9%)	60 (15.7%)	321 (84.3)	<0.001
	>64 years	121 (24.1%)	59 (48.8%)	62 (51.2%)	
Nationality	Italian	491 (97.8%)	121 (24.6%)	370 (75.4%)	0.310
	Other	11 (2.2%)	1 (9.1%)	10 (90.9%)	
Educational level	Primary school or less	52 (10.4%)	22 (42.3%)	30 (57.7%)	0.006
	Lower secondary school	209 (41.6%)	53 (25.4%)	156 (74.6%)	
	High school	194 (38.6%)	37 (19.1%)	157 (80.9%)	
	Bachelor’s degree or higher	47 (9.4%)	10 (21.3%)	37 (78.7%)	
Health literacy (HLS-EU-Q6)	Inadequate	35 (7%)	11 (31.4%)	24 (68.6%)	0.518
	Problematic	143 (28.5%)	32 (22.4%)	111 (77.6%)	
	Sufficient	255 (50.8%)	60 (23.5%)	195 (76.5%)	
	Missing	69 (13.7%)	-	-	
Employment	No one	302 (60.2%)	89 (29.5%)	213 (70.5%)	0.006
	Non-public employment	76 (15.1%)	15 (19.7%)	61 (80.3%)	
	Public employment	112 (22.3%)	15 (13.4%)	97 (86.6%)	
	Health or social-health worker	12 (2.4%)	3 (25%)	9 (75%)	
Living with people >64 years old or with people with chronic diseases		149 (29.7%)	40 (26.8%)	109 (73.2%)	0.426
Risk conditions or diseases	Diabetes	23 (4.6%)	12 (52.2%)	11 (47.8%)	0.003
	Obesity	35 (7%)	11 (31.4%)	24 (68.6%)	0.414
	Heart diseases	25 (5%)	15 (60%)	10 (40%)	<0.001
	Pulmonary diseases	17 (3.4%)	9 (52.9%)	8 (47.1%)	0.009
	Diseases of the immune system	13 (2.6%)	2 (15.4%)	11 (84.6%)	0.537
	Chronic kidney diseases	4 (0.8%)	2 (50%)	2 (50%)	0.250
	Chronic liver diseases	2 (0.4%)	2 (100%)	0	0.059
	Organ or bone marrow transplant	1 (0.2%)	1 (100%)	0	0.243
	Chronic neurological diseases	9 (1.8%)	3 (33.3%)	6 (66.7%)	0.695
	Oncological diseases (prior 5 years)	10 (2%)	6 (60%)	4 (40%)	0.016
	Hematological diseases	2 (0.4%)	0	2 (100%)	1.000
	Pregnancy	3 (0.6%)	0	3 (100%)	0.581
	Surgery under general anesthesia (prior year)	23 (4.6%)	10 (43.5%)	13 (56.5%)	0.043
	At least one of the previously listed	134 (26.7%)	52 (38.8%)	82 (61.2%)	<0.001
Smoking habits	Never smokers	254 (50.6%)	60 (23.5%)	194 (76.4%)	0.576
	Current smokers, fewer than 10 cigarettes/day	65 (12.9%)	12 (18.5%)	53 (81.5%)	
	Current smokers, 10–20 cigarettes/day	74 (14.7%)	20 (27%)	54 (73%)	
	Current smokers, more than 20 cigarettes/day	7 (1.4%)	1 (14.3%)	6 (85.7)	
	Former smokers	102 (20.3%)	29 (28.4%)	73 (71.6%)	

**Table 2 ijerph-19-06688-t002:** Descriptive analysis of the numerical variables for the whole sample and by influenza vaccination uptake in the 2019–2020 season. NOTE: * Mann–Whitney test for independent samples.

Variables		Mean (SD); Median (IQR)	Influenza Vaccination		
			Yes	No/Don’t Remember	*p* *
			Mean (SD); Median (IQR)	Mean (SD); Median (IQR)	
Age		49.5 (17.9); 53 (33–64)	59.4 (16.0); 65 (52.7–69.2)	46.3 (17.3); 48 (30–61)	<0.001
Age, excluding >64 years old		42.8 (15.1); 46 (28–57)	47.4 (14.8); 52.5 (34–59)	41.9 (15.0); 44 (27–55)	0.007
HL score (HLS-EU-Q6)		2.9 (0.6); 3 (2.7–3.3)	2.9 (0.57); 3 (2.6–3.17)	3.0 (0.59); 3 (2.7–3.3)	0.245
Living conditions	N of cohabitants	3 (1.3); 3 (2–4)	2.6 (1.35); 2 (2–3)	3.09 (1.26); 3 (2–4)	<0.001
	N of rooms in the house (1 missing)	4.7 (2.3); 4 (4–5)	4.5 (1.65); 4 (4–5)	4.8 (2.45); 4 (4–5)	0.478
	N of bedrooms (1 missing)	2.3 (0.8); 2 (2–3)	2.3 (0.86); 2 (2–3)	2.3 (0.78); 2 (2–3)	0.579

**Table 3 ijerph-19-06688-t003:** Multivariate logistic regression final models for predicting vaccination uptake in the 2019–2020 season (odds ratio of having taken the vaccine). A: whole sample; B: people ≤64 years old; C: people >64 years old; D: people with sufficient health literacy (HL) according to HLS-EU-Q6; E: people with problematic or inadequate health literacy (HL) according to HLS-EU-Q6. NI = not included.

Variables	A: Whole Sample N = 502 (24.3% Vaccinated)		B: ≤64 Years N = 381 (15.7% Vaccinated)		C: >64 Years N = 121 (48.8% Vaccinated)		D: Sufficient HL N = 178 (24.1% Vaccinated)		E: Problematic or Inadequate HL N = 255 (23.5% Vaccinated)	
	OR [95% CI]	*p*	OR [95% CI]	*p*	OR [95% CI]	*p*	OR [95% CI]	*p*	OR [95% CI]	*p*
Age (continuous)	1.05 [1.03–1.07]	<0.001	1.02 [1.00–1.04]	0.039	NI	-	1.04 [1.02–1.07]	<0.001	1.05 [1.02–1.08]	<0.001
Diabetes	NI	-	4.09 [1.15–14.51]	0.029	NI	-	NI	-	6.34 [1.70–23.58]	0.006
Heart diseases	2.98 [1.24–7.19]	0.015	4.19 [1.18–14.7]	0.027	NI	-	NI	-	4.53 [1.15–17.80]	0.030
Pulmonary diseases	6.18 [2.01–19.04]	0.002	5.84 [1.89–18.05]	0.002	NI	-	NI	-	8.73 [2.11–36.12]	0.003
Oncological diseases (prior 5 years)	NI	-	5.30 [1.12–25.04]	0.035	NI	-	NI	-	NI	-
Surgery under general anesthesia (prior year)	3.14 [1.23–8.06]	0.017	NI	-	NI	-	NI	-	6.38 [1.71–23.76]	0.006
At least one of the listed risk condition or diseases *	NI	-	NI	-	3.22 [1.44–7.23]	0.004	NI	-	NI	-
Educational level	NI	-	NI	-	NI	-	NI	-	NI	
Employment	NI	-	NI	-	NI	-	NI	-	NI	
Number of cohabitants	NI	-	NI	-	NI	-	NI	-	NI	

* diabetes, obesity, heart diseases, pulmonary diseases, diseases of the immune system, chronic kidney diseases, chronic liver diseases, organ or bone marrow transplant, chronic neurological diseases, oncological diseases (prior 5 years), hematological diseases, pregnancy, surgery under general anesthesia (prior year).

## Data Availability

The dataset generated and analyzed during the current study is available from the corresponding author on reasonable request.

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
