# Peer review of "Predictors of Influenza Vaccination Uptake and the Role of Health Literacy among Health and Social Care Volunteers in the Province of Prato (Italy)"

_ijerph, 2022, doi:10.3390/ijerph19116688_

Round 1

Reviewer 1 Report

In this manuscript authors performed a cross-sectional survey to determined the predictors of influenza vaccination in Prato, Italy during the pandemic. They also evaluated the role of health literacy on influenza vaccine uptake in health and social care volunteers. This study is well performed and explained clearly. My minor suggestions are as follows:

  1. provide full form of HL in abstract.
  2. as there is overall low influenza vaccine uptake during this study, provide the previous season influenza vaccine histories in the area. 
  3. was the impact of ongoing COVID pandemic considered on influenza vaccine uptake? 

Reviewer 2 Report

  1. The article presents an interesting study. The article is interesting and timely but it suffers from various limitations that must be addressed before it is accepted for publication. I recommend a major revision.
  2. In the introduction, the authors should clearly present the following: What is missing and what is the gap? Why there is a need to conduct this study? Who will benefit? What is the novelty of this work?  What are its main contributions What is the underlying theory? All these questions are unclear to me in the current version of the article. 
  3. Authors have cited too many old social media studies. Please replace those with more recent studies. 
  4. The discussion is difficult to follow. Please separate implications from the discussion. The new section should have two sub-sections - Theoretical and practical implications. Limitations and future work. 

Reviewer 3 Report

- TITLE

o The title is too long

- METHODOLOGY

o No measures were specified to reduce potential sources of bias.

o No sample size was determined. It would have been interesting to compare the characteristics of the sample and the characteristics of those who responded to the questionnaire.

- RESULTS

o Is the population of origin also characterized by having 65% men?

o It would be advisable to compare the population to which the questionnaire was administered versus those who responded. It would also be interesting to make a sample size calculation to see if the number of individuals who responded was sufficient to make an inference.

o It would be interesting to reflect the response rate to the questionnaire.

- DISCUSSION

o They claim that the individuals who responded to the questionnaire are representative and therefore make some inferences. No data?

- CONCLUSIONS.

o The last conclusion would be more from the discussion than a conclusion drawn from the results obtained.

  • REFERENCES

o The bibliographic references that have links are poorly referenced. They are missing [internet] [accessed on......]

o The citations are not very current. It is recommended to look for articles from 2021 and 2022.

Round 2

Reviewer 3 Report

Of the previous suggestions that were made, the following have not yet been made:

·       No sample size was determined. It would have been interesting to compare the characteristics of the sample and the characteristics of those who responded to the questionnaire.

·       It would be advisable to compare the population to which the questionnaire was administered versus those who responded. It would also be interesting to make a sample size calculation to see if the number of individuals who responded was sufficient to make an inference

·       The bibliographic references that have links are poorly referenced. They are missing [internet] [accessed on......]
